

# OpenIFS/AC: atmospheric chemistry and aerosol in OpenIFS 43r3

Vincent Huijnen[1], Philippe Le Sager[1], Marcus O. Köhler[2], Glenn Carver[2], Samuel Rémy[3], Johannes Flemming[2], Simon Chabrillat[4], Quentin Errera[4], Twan van Noije[1]

[1]Royal Netherlands Meteorological Institute, De Bilt, The Netherlands
[2]ECMWF, Shinfield Park, Reading, UK
[3]HYGEOS, Lille, France
[4]Royal Belgian Institute for Space Aeronomy, BIRA-IASB, 1080 Brussels, Belgium

*Correspondence to*: Vincent Huijnen (Vincent.Huijnen@knmi.nl)

**Abstract.** In this paper, we report on the first implementation of atmospheric chemistry and aerosol as part of the European Centre for Medium-Range Weather Forecasts (ECMWF) OpenIFS model. For this, modules and input data for model cycle CY43R3, which have been developed as part of the Copernicus Atmosphere Monitoring Service (CAMS), have been ported to OpenIFS: the modified CB05 tropospheric chemistry scheme, the AER bulk-bin tropospheric aerosol module, as well as the option to use BASCOE-based stratospheric ozone chemistry. We give an overview of the model, and describe the datasets used for emissions and dry deposition, which are similar to those used in the model configuration applied to create the CAMS Reanalysis. We evaluate two reference model configurations with and without the stratospheric chemistry extension against standard observational datasets for tropospheric ozone, surface carbon monoxide (CO), tropospheric nitrogen dioxide ($NO_2$) and aerosol optical depth. The results give basic confidence in the model implementation and configuration. This OpenIFS version with atmospheric composition components is open to the scientific user community under a standard OpenIFS license.

## 1 Introduction

The presence and amounts of reactive trace gases and aerosol in the atmosphere is important for local air quality conditions (Im et al., 2018), as well as for the Earth's radiation balance (IPCC Chapter 6, 2021). Hence, knowledge with regard to their variability and evolution helps to identify policy measures aimed at improving air quality or mitigating near-term climate change, or both. Furthermore, good understanding, and in turn constraints, on atmospheric composition are important building blocks in the generation of satellite retrieval products for many species and trace gases (van Geffen et al., 2021; De Smedt et al., 2018).

For many of these applications, chemistry transport models (CTMs), such as the global model TM5-MP (Williams et al., 2017; Huijnen et al., 2010), are appropriate. However, limitations exist in the sense that these require meteorological input from a separate datastream, which implies that any feedback of the chemistry and aerosol simulation on the meteorology is either ignored or can only be accounted for through an external coupling with the meteorological model. With chemistry and



aerosol components embedded in the meteorological model such an external coupling is no longer needed. This makes the model both computationally more efficient and more consistent in terms of transport. It also allows for a full description of feedbacks between composition and meteorology. One limitation is that the atmospheric composition module needs to be run at the same grid as the meteorological model. Therefore the resolution of an OpenIFS model configuration without atmospheric

composition typically has to be coarsened to achieve acceptable computational costs when switching on this module.

In the framework of the Copernicus Atmosphere Monitoring Service (CAMS, https://atmosphere.copernicus.eu/) the tropospheric chemistry module originating from TM5 based on a modified version of the Carbon Bond 2005 (CB05) scheme (Yarwood et al., 2005) has been implemented into ECMWF's Integrated Forecasting System (IFS) (Flemming et al., 2015). This module has been extended and updated over time (Huijnen et al., 2017; Williams et al., 2021). Alternative chemistry

modules are also supported (Huijnen et al., 2017), particularly a module for stratospheric chemistry based on BASCOE chemistry, which can be switched on in combination with the CB05-type chemistry (Huijnen et al., 2016). Likewise a module describing the fate of aerosol has been developed over time (Morcrette et al., 2009; Rémy et al., 2017, 2021a). The composition aspects have also been integrated in ECMWF's data assimilation system (Inness et al., 2015) allowing the assimilation of satellite observations of trace gases and aerosol. This enables running the model in an operational context to provide global

analyses, reanalyses and forecasts of atmospheric composition, and may in general be referred to as IFS-Composition (or IFS-CB05-AER, IFS-CB05-BASCOE), etc., to specify a particular configuration).

The OpenIFS model (https://confluence.ecmwf.int/display/OIFS) is a portable version of the ECMWF global IFS numerical weather prediction model that is available to research institutes and universities. It enables the scientific community to use

selected cycles of the ECMWF operational IFS model on local computers at their own institutes. OpenIFS has the same forecasting capability as IFS, it supports the same grid resolutions up to the operational resolution, and it can be used for ensemble forecasting (Ollinaho et al., 2021). The model codes differ however as the IFS code for data assimilation and observation processing have been removed from OpenIFS. OpenIFS is used in professional training and university teaching (e.g., Szépszó et al., 2019), as well as a research tool to investigate atmospheric processes on a wide range of topics from NWP

to climatologically relevant time scales (https://confluence.ecmwf.int/display/OIFS/OpenIFS+publications). In the realm of climate applications, OpenIFS will form the atmospheric component of the next generation (version 4) of the European community Earth system model. A recent, more detailed description of OpenIFS 40r1 is available in Sparrow et al. (2021). The current version, OpenIFS 43r3 (version 2), that is used in this work is based on IFS cycle CY43R3 which is described in detail both at the scientific (ECMWF, 2017a, 2017b) and technical levels (ECMWF, 2017c, 2017d). Until now OpenIFS did

not contain support for atmospheric composition modeling as has been developed in CAMS.

Here we present a first version of OpenIFS that contains modules for atmospheric composition, based on the CB05, CB05-BASCOE and AER, which we refer to as OpenIFS/AC. This version relies on model code that has been developed as part of CAMS, although alternative schemes will be introduced in future versions of the model. In particular, a modal aerosol



representation based on M7 (Vignati et al., 2004) is currently in development for use in EC-Earth4. Thus, in contrast to the various configurations of EC-Earth3 (Döscher et al., 2021), in which aerosol is either prescribed or simulated interactively using a two-way coupling between IFS and TM5 (van Noije et al., 2021), the use of OpenIFS/AC in EC-Earth4 will enable simulating aerosol and optionally tropospheric and/or stratospheric chemistry using tracers inside OpenIFS.

Apart from the actual model code, which involves parameterizations of processes affecting the budgets of atmospheric tracers (emission, chemical conversion, transport, deposition), the integration of composition modeling also includes the preparation of required input data (tracer definition, emissions, deposition specifications), along with elemental post-processing (data archiving, and validation). In the following subsections we provide an overview of the parameterizations and the corresponding input datasets, required to perform model simulations with OpenIFS/AC. Furthermore we provide a first, global evaluation of the model performance in terms of reactive trace gases and aerosol. The purpose of this evaluation is primarily to present the technical feasibility of the system, and provide a benchmark for future developments. The version of the model as presented here is made available to the community under the OpenIFS license. In Sect. 8 we provide information that describes the access to the model code, along with its elemental input data and pre- and post-processing scripts that are necessary to run the model as described here.

## 2 OpenIFS meteorology and model configurations

OpenIFS 43r3 is based on IFS cycle CY43R3, which was ECMWF's operational model configuration in the time period July 2017 – June 2018. The IFS is a spectral NWP model that applies the semi-Lagrangian (SL) semi-implicit method to solve the governing dynamical equations. The simulation of the hydrological cycle is described in Forbes et al. (2011) and includes prognostic representations of cloud fraction, cloud liquid water, cloud ice, rain and snow.

As described in Flemming et al. (2015), the tracer transport is modeled by advection of the tracer mass mixing ratios by the SL method (Temperton et al., 2001; Hortal, 2002). Vertical redistribution by diffusion and convection is described in Beljaars and Viterbo (1998) and Bechtold et al. (2014). Emission and dry deposition are handled as part of the diffusion scheme.

The chemical trace gases and aerosol, and their processes are represented only in grid-point space using mass mixing ratio as the prognostic variable.

In our standard configuration the horizontal grid is a reduced Gaussian grid (Hortal and Simmons, 1991). As the IFS, OpenIFS can be run at varying vertical and horizontal resolutions. A standard horizontal resolution for which input data is available, is the $T_L255$ spectral resolution, which corresponds to a grid box size of about 80 km. In our current configuration the vertical discretisation uses 91 levels up to the model top at 0.01 hPa (80 km) using a hybrid sigma-pressure coordinate. The vertical extent of the lowest level is about 10 m; it is 90 m at about 300 m above ground, and approximately 400 m at about 10 km in height.

Surface fluxes of energy and water over land, and the corresponding sub-surface quantities, are represented in OpenIFS with the Tiled ECMWF Scheme for Surface Exchanges over Land (HTESSEL, Balsamo et al., 2009). Recent improvements for



OpenIFS 43r3 include a revised land surface hydrology which addresses surface runoff. This also includes a new formulation to represent inland water bodies such as resolved lakes and sub-grid coastal water, using the Fresh-water Lake model Flake. A technical description of the land surface scheme is available in ECMWF (2017b). OpenIFS further includes two-way coupling to the sea surface via an ocean surface wave model (ECMWF, 2017d).

This version of OpenIFS uses the modular ecRad radiation scheme (Hogan and Bozzo, 2018) which uses the Monte Carlo Independent Column Approximation (McICA) code. In comparison to the earlier McRad scheme, ecRad uses a modular approach and results in longwave radiation transfer improvements, reduced temperature profile biases, and less noise in partially cloudy conditions.

OpenIFS is designed originally as a forecast model in a NWP framework. However, various constraints can be provided to the model as indicated in Table 1 through specifying the initial conditions and composition surface fluxes at the start of the run, and through different ways to constrain meteorology and surface fluxes throughout the run. This allows use of the model for different application types. Here we present the use of OpenIFS/AC primarily in the 'nudging' configuration. In this configuration selected meteorological quantities (e.g. winds and surface pressure) are nudged towards prescribed input data with specified relaxation time constants. In this configuration the emissions and deposition velocities are standard updated daily from a prepared dataset. This allows running the model efficiently for longer periods (typically a year), while keeping the meteorology and emissions constrained towards external datasets on a high temporal and spatial resolution.

Nevertheless, we develop the model such that it can be equally used in other configurations. A 'cyclic forecast' configuration is designed to execute free forecast runs repetitively, using updated initialization of meteorology and emissions from an external source, such as ECMWF operational meteorological analyses. In this configuration no nudging, or surface flux updates, within each forecast is applied. This is in principle suitable for near-real time applications, e.g., to provide a-priori model composition fields for satellite retrieval products.

When extending the forecast range of such an individual forecast run, while not enforcing any constraints (nudging) of meteorology, this results in a 'free run' of both meteorology and composition. This configuration therefore allows to study the model climate, and could be used to study composition-meteorology feedbacks. It is expected to be best suited for climate applications, e.g., within an EC-Earth framework. Still specific nudging, e.g. for sea surface temperature, as well as using updates of emissions throughout the run to account for seasonal to decadal changes, would be necessary to make this configuration most useful for scientific purposes. Indeed Table 1 only reflects the reference model configurations, while in practice mixtures may turn out more useful.

Also an important consideration in this context is the availability of meteorological input data at the actual resolution of the model. If the user chooses to run the model with a (horizontal/vertical) grid resolution that differs from that of the input data then a re-gridding procedure is required.



**Table 1.** Overview of standard configurations to operate OpenIFS/AC experiments.

| Configuration | Meteorology and composition initialization specifications | Meteorological nudging and emissions specification | Application |
|---|---|---|---|
| Cyclic forecast | Daily initialization of meteorology from external source (analysis), composition from preceding forecast or external source | No relaxation of meteorology; updated emissions at the start of each forecast | Near-real time composition modeling; short experimenting on composition modeling. Not suitable for composition-meteorology feedback |
| Nudging | Initialization of meteorology and composition only at first start; use of restart fields | Meteorology relaxation towards user specified fields from external source; daily emission specification | Extended runs with specified meteorology, for composition modeling. Limited applicability for composition-meteorology feedback studies. |
| Free run | Initialization of meteorology and composition only at first start; use of restart fields | No relaxation for meteorology; use of monthly emissions | Model climate experiments. Also allows investigation of composition-meteorology feedback |

## 3 Atmospheric chemistry and aerosol

In this section we describe key aspects concerning the atmospheric chemistry and aerosol modules, and the main options
available in this version of OpenIFS/AC. For the troposphere this concerns the modified CB05 mechanism (see Sect. 3.1),
while for the stratosphere one may choose between the linear ozone model or the BASCOE-based module for stratospheric
ozone chemistry (Sect. 3.2). A description of the AER bulk-bin tropospheric aerosol module is given in Sect. 3.3. The
specification of reference emissions, and dry and wet deposition is given in Sect. 3.4 and 3.5, respectively. Some comments
on tracer advection aspects in OpenIFS are given in Sec 3.6.

## 3.1 Tropospheric chemistry

The modified CB05 module for tropospheric chemistry in our version of OpenIFS/AC is based on the CB05 scheme from
Yarwood et al. (2005). It uses a lumping approach for organic species depending on their functional groups. In its application
in the TM5 and CAMS models, modifications and extensions have been developed which include an explicit treatment of C1
to C3 species (Williams et al., 2013), as well as parameterization of $SO_2$, dimethyl sulfide (DMS), methane sulfonic acid
(MSA) and ammonia ($NH_3$) chemistry (see also Huijnen et al., 2010). Gas–aerosol partitioning of nitrate and ammonium is





calculated using the Equilibrium Simplified Aerosol Model (EQSAM; Metzger et al., 2002). The modified band approach (MBA) is adopted for the online computation of photolysis rates in the troposphere (Williams et al., 2012) and uses seven absorption bands across the spectral range 202–695 nm, accounting for cloud and aerosol optical properties. Heterogeneous

reactions and photolysis rates in the troposphere depend on cloud droplets as well as prognostic aerosol tracers. The reaction rates for the troposphere follow the recommendations given in either Jet Propulsion Laboratory (JPL) evaluation 17 (Sander et al., 2011) or Atkinson et al. (2006).

The complete chemical mechanism as applied for the troposphere is referred to as "tc01a", and consists of 55 tracers, 104 gas-phase reactions, 20 photolysis rates, 3 heterogeneous reactions and 2 aqueous phase reactions. It is extensively documented in

Flemming et al. (2015).

Two solvers are available, either based on the Euler backward iterative (EBI) methode, or based on Kinetic PreProcessor (KPP) routines, using the four stages and third-order Rosenbrock solver (Sandu and Sander, 2006).

## 3.2 Stratospheric chemistry

Above the tropopause stratospheric ozone is either governed by a linear ozone chemistry scheme (Cariolle and Teyssèdre, 2007), or by explicit modeling of stratospheric composition. For this last option, the chemical scheme and the parameterisation for polar stratospheric clouds (PSCs) from the BASCOE system version "sb14a" have been implemented in the IFS in the framework of CAMS (Huijnen et al., 2016), and here made available for OpenIFS/AC. Photolysis rates were computed offline by the TUV package (Madronich and Flocke, 1999), and are provided as lookup tables as a function of log-pressure altitude,

ozone overhead column and solar zenith angle.

Photolysis rates for reactions occurring in both the troposphere and stratosphere are merged at the interface in order to ensure a smooth transition between the two schemes. To distinguish between the tropospheric and stratospheric regime, we use a chemical definition of the tropopause level, whereby tropospheric grid cells are defined at $O_3 < 200$ ppb and $CO > 40$ ppb for $p > 40$ hPa. Gas-phase and heterogeneous reaction rates are taken from JPL evaluation 17 (Sander et al., 2011) and JPL

evaluation 13 (Sander et al., 2000), respectively. The reaction mechanism in the stratosphere is solved using a KPP-based four stages and third-order Rosenbrock solver (Sandu and Sander, 2006).

## 3.3 Tropospheric aerosol

The standard aerosol module as available in OpenIFS is based on the AER module as developed in CAMS (Morcrette et al.,

2009 Rémy et al., 2019), as of the status of CY43R3. It consists of a bulk–bin scheme, originally derived from the LOA/LMDZ model (Boucher et al., 2002; Reddy et al., 2005), with aerosol species characteristics as shown in Table 2. The prognostic species are sea salt, desert dust, organic matter (OM), black carbon (BC), and sulfate. Optionally, the OpenIFS/AC aerosol module can be run in stand-alone mode (without interaction with chemistry), in combination with one gas-phase precursor:





SO$_2$. The SO$_2$ to sulfate conversion rate is parameterized as a function of relative humidity and temperature. In our
configuration we run the module coupled with CB05-based tropospheric chemistry.

Both sea salt and desert dust are represented with three bins. As described in Reddy et al. (2005), sea salt emissions as well as
sea salt particle radii are expressed at 80 % relative humidity (RH). This is different from all the other aerosol species in AER,
which are expressed as dry mixing ratios (0 % RH). Users should pay special attention to this when dealing with a diagnosed
sea salt aerosol mass mixing ratio, which needs to be divided by a factor of 4.3 to convert to the dry mass mixing ratio in order
to account for hygroscopic growth and change in density. For both dust and sea salt, there is no mass transfer between bins.
For OM and BC the AER module accounts for both hydrophilic and hydrophobic fractions. Here an ageing process describes
mass transfer from the hydrophobic to hydrophilic OM and BC.

Ageing of hydrophobic OM and BC into hydrophilic aerosol is modeled through a fixed lifetime of 1.16 days (Rémy et al.,
2019). Hygroscopic growth of aerosol which is strongly affecting the optical properties is implicitly modeled through a growth
factor depending on the ambient relative humidity.

A parameterization for (coarse- and fine-mode) nitrates and ammonium aerosol with production from the chemistry is available
but not applied here. In all, AER is thus composed of 11 prognostic aerosol variables, or 14 when including nitrates and
ammonium, representing tropospheric aerosol. Note that later cycles of IFS, as operated in CAMS, also include improved
representation of secondary inorganic and  organic aerosol, together with various other updates (Rémy et al., 2019; Rémy et
al., 2021). This is expected to become available in future versions of OpenIFS/AC.

**Table 2.** Aerosol species and parameters of the size distribution associated with each aerosol type in OpenIFS/AC ($r_{mod}$: mode
radius, $\rho$: particle density, $\sigma$: geometric standard deviation). Values are for the dry aerosol apart from sea salt, which is given
at 80 % relative humidity (RH).

| Aerosol type | Size bin limits (sphere radius; μm) | $\rho$ [kg m$^{-3}$] | $r_{mod}$ [μm] | $\sigma$ |
|---|---|---|---|---|
| Sea salt (80% RH) | 0.03-0.5 0.5-5.0 5.0-20 | 1183 | 0.1992, 1.992 | 1.9, 2.0 |
| Desert dust | 0.03-0.55 0.55-0.9 0.9-20 | 2610 | 0.29 | 2.0 |
| Black carbon | 0.005-0.5 | 1000 | 0.0118 | 2.0 |
| Sulfates | 0.005-20 | 1760 | 0.0355 | 2.0 |
| Organic matter | 0.005-20 | 2000 | 0.021 | 2.24 |




### 3.4 Emissions and surface boundary conditions

Application of trace gas and aerosol emissions in OpenIFS/AC are provided through specific GRIB-files which contain the total daily or monthly emissions per tracer, as a combination of anthropogenic, biogenic, soil, oceanic and biomass burning

sources. Furthermore, the aerosol biomass burning emissions are treated separately. The surface emissions are injected as lower boundary flux in the diffusion scheme. Additionally, atmospheric emissions of aircraft $NO_x$ are prescribed, while lightning $NO_x$ is parameterised depending on the convection as described in Flemming et al., (2015).

In our current configuration as presented here we use MACCity anthropogenic trace gas and aerosol emissions (Granier et al., 2011), with upscaled wintertime CO traffic emissions according to Stein et al. (2014). Following Rémy et al. (2019), the

aerosol the black carbon emissions are distributed by 20 % into the hydrophilic and the remaining 80 % into the hydrophobic black carbon tracers as in Reddy et al. (2005). The MACCity emissions of organic carbon are translated into 26 Tg yr$^{-1}$ OM emissions using an OM:OC ratio of 1.8. In the model the OM emissions are divided evenly between hydrophilic and hydrophobic OM tracers. In this configuration the secondary organic aerosol (SOA) is added as part of the organic matter species and is emitted at the surface and contributes with biogenic (19.1 Tg yr$^{-1}$) and anthropogenic (144 Tg yr$^{-1}$) sources

following Dentener et al., (2006) and Spracklen et al. (2011), see also Rémy et al., (2019). Although this parameterization has been shown to be beneficial in addressing negative biases in AOD climatology, the simplistic treatment of direct emissions currently also results in positive model biases for surface PM.

Biogenic emissions originate from the MEGAN-MACC inventory (Sindelarova et al., 2014), while oceanic emissions are taken from POET-based oceanic emissions (Granier et al., 2005). This is consistent with the emissions used for the CAMS

reanalysis (Inness et al., 2019).

Daily biomass burning emissions of trace gases as well as OM and BC are taken from the Global Fire Assimilation System (GFAS) version 1.2, which uses satellite retrievals of fire radiative power (Kaiser et al., 2012). For the aerosol tracers a scaling factor of 3.4 is applied to the GFAS biomass burning sources when used in the IFS. This factor has been introduced to minimise the error compared to MODIS AOD (Kaiser et al., 2012, Rémy et al., 2019), and may reflect unrepresented precursors of OM

and BC. Biomass burning emissions of trace gases are so far by default released at the surface, while for aerosol daily-specific injection heights are adopted as provided along with the GFAS fire emissions, see also Rémy et al., (2017).

The actual emission totals used in the simulation for 2010 are given in Table 3.




**Table 3.** Overview of standard emissions used in the current OpenIFS/AC configuration (Tg species yr$^{-1}$ unless specified otherwise)

| Tracer | Anthropogenic | Biogenic | Biomass burning | Other |
|---|---|---|---|---|
| CO | 599 | 93 | 325 | 20 (oceanic) |
| NO[a] | 71 | - | 9.5 | 10.5 (soil) |
| $CH_2O$ | 3.4 | 4.9 | 5.6 | - |
| $CH_3OH$ | 2.2 | 134 | 14.4 | - |
| $C_2H_6$ | 3.3 | 0.3 | 2.2 | 1.0 (oceanic) |
| $C_2H_5OH$ | 3.2 | 19.6 | 0.2 | - |
| $C_2H_4$ | 7.6 | 30.5 | 4.7 | 1.4 (oceanic) |
| $C_3H_8$ | 4.0 | 0.03 | 1.4 | 1.3 (oceanic) |
| $C_3H_6$ | 3.5 | 15.5 | 2.9 | 1.5 (oceanic) |
| Paraffins (Tg C) | 31 | 1 | 0.5 | - |
| Olefins (Tg C) | 2.4 | 0.7 | 0.6 | - |
| Aldehydes (Tg C) | 1.1 | 6.5 | 2.6 | |
| $CH_3COCH_3$ | 1.3 | 38 | 3.8 | - |
| Isoprene | - | 597 | - | - |
| Terpenes | - | 98 | - | - |
| $SO_2$ | 97 | - | 0.8 | 13 (volcanic) |
| DMS | - | - | - | 38 (oceanic) |
| $NH_3$ | 42 | 2 | 11.3 | 8 (soil) |
| OM (Tg OM) | 190 | - | 68 | |
| BC | 5.1 | | 6.9 | |

[a]In addition to the emissions specified here, also 0.8 Tg N yr−1 aircraft emissions (Lamarque et al.; 2010) and 4.1 Tg N yr$^{-1}$ lightning NOx emissions are applied.






Sea salt emissions in this version of OpenIFS/AC-AER follow the Monahan et al. (1986) parameterization as described in Morcrette et al. (2009). Also for desert dust emissions the parameterization as developed in Morcrette et al. (2009) is adopted, based on Ginoux et al. (2001), and as explained also in Rémy et al. (2019).

Methane ($CH_4$), as well as $N_2O$ and a selection of chlorofluorocarbons (CFCs), in the case of running with BASCOE-based stratospheric chemistry, are prescribed at the surface as boundary conditions. While for $N_2O$ and CFC annually and zonally fixed values are currently assumed (Huijnen et al., 2016b), for $CH_4$ zonally and seasonally varying surface concentrations are adopted based on a climatology derived from NOAA flask observations ranging from 2003 to 2014.

**3.5 Deposition and sedimentation**

In the current configuration of OpenIFS/AC dry deposition velocities for trace gases are provided as trace gas specific monthly mean fields from a simulation using the approach discussed in Michou et al. (2004). A diurnal variation is applied to these deposition velocities, described by a cosine function of the solar zenith angle with ±50 % variation. We note that more recent versions of the IFS as operated in CAMS use an online computation of dry deposition velocities. Wet scavenging, including
in-cloud and below-cloud scavenging as well as re-evaporation, is treated following Jacob et al. (2000). The reader is referred to Flemming et al. (2015) for further details on the dry and wet deposition parameterisations for the gases.

Aerosol dry and wet deposition and aerosol sedimentation follow the implementation proposed by Morcrette et al (2009), as also described by Rémy et al. (2019). More specifically, dry deposition is modelled following the Reddy et al. (2005)
parameterization, with fixed deposition velocity values per species, which are different over continents and ocean for sea salt and sulfate aerosol. Wet deposition includes parameterizations for in-cloud and below-cloud scavenging, and make use of the cloud water and precipitation fluxes in the IFS. All aerosol tracers except hydrophobic OM and BC are subject to wet deposition. Sedimentation is described according to the Thomkins et al. (2005) parameterization, and is only applied to super-coarse dust and sea salt.


**3.6 Tracer mass advection**

The SL advection scheme does not conserve mass, mainly due to errors associated with the interpolation method to compute the start point of the trajectory of the variable towards each individual grid point due to advection. Therefore a mass fixer needs to be applied to ensure global mass conservation (Diamantakis and Flemming, 2014). If not properly handled, this may result
in various unwanted artefacts, such as spurious drifts on the troposphere for long-lived tracers such as $CH_4$ and $CO_2$ (Agusti-Panareda et al., 2017), mass redistribution and drifts in stratospheric composition (Huijnen et al., 2016), and local plume distortion (Diamantakis and Flemming, 2014). While these aspects are acknowledged and subject of model improvement in



more recent cycles of the IFS, various options already exist in OpenIFS 43r3v2, including specific modifications for tracers representing reactive trace gases in order to optimise their mass conservation properties.

First, the user may choose the interpolation method to be quasi-monotonic to avoid under- or overshoots together with negative values. Secondly, the user may choose between various mass fixer algorithms. For reactive trace gases and aerosol the preferred option is to use a mass fixer that is proportional to the amount of tracer mass within each grid cell. For long-lived tracers ($CH_4$ and $CO_2$) it was pointed out that the Bermejo & Conde scheme is prefered (Agusti-Panareda et al., 2017).

Finally, the use of family tracer advection has been introduced in the context of stratospheric chemistry (Huijnen et al., 2016).

The $NO_y$ (= $NO + NO_2 + NO_3 + HNO_3 + HO_2NO_2 + 2 \times N_2O_5 + ClNO_2 + ClONO_2 + BrONO_2$), $Cl_y$ (=$2 \times Cl_2O_2 + OClO + BrCl + HOCl + ClONO_2 + Cl + HCl + ClO + ClNO_2 + 2 \times Cl_2 + ClOO$) and $Br_y$ ( $BrCl$, $HOBr$, $BrONO_2$, $Br$, $HBr$, and $BrO$)  families have much smoother spatial gradients, especially near the terminator where large gradients exist for trace gases that are subject to photolytic production or loss, and therefore do not require a mass fixing. Advection of these family tracers hence results in much smaller mass conservation errors compared to that of individual trace gases. The local mass of individual trace gases is

then computed by application of the same partitioning ratios as before the advection steps, hence advection is assumed to conserve the same partitioning within the advected families.

## 5 Input and output data, and its handling

In this section we describe some specifics regarding input and output data for OpenIFS/AC, to the extent this is different compared to a standard OpenIFS configuration. Like the IFS, OpenIFS produces GRIB-fields containing 2D and 3D output

fields with essentially user-specific quantities. Also the use of the XIOS infrastructure (Yepes-Arbós et al., 2022) to allow NetCDF output is supported. Pre- and post-processing scripts have been developed to handle the model input and output, of which a basic selection is available in the distribution package.

### 5.1 Input data

The following sets of atmospheric composition input data are required to run OpenIFS/AC.

1.    Trace gas initial conditions (IC): 3D fields that are integral part of the `ICMGG${expid}INIUA` input datafile in GRIB format. In our experiments these fields are taken from existing IFS-Composition experiments as run in the CAMS configuration, but they may equally be provided through other sources. These tracer fields are defined in OpenIFS/AC by their namelist entries `YCHEM_NL(1:NCHEM)`, where the tracer specifics (GRIB number, some

physical parameters, together with transport,  deposition and mass fixer settings) are configured. We note that for various tracers (particularly those required to run stratospheric chemistry) the official GRIB number in table 217 is not yet supported in this cycle of OpenIFS. Here we use entries from the free, experimental GRIB table 216 instead, and rename the GRIB numbers when using a more recent CAMS experiment to provide input data.





2. Aerosol IC: similar to trace gas IC these are provided in `ICMGG${expid}INIUA` datafile, with tracer fields defined in the namelist entries `YAERO_NL(1:NACTAERO)`. Table files define the list of chemistry and aerosol tracers, together with their main tracer specifics. A script is used to generate the corresponding namelist entries.

3. Trace gas and aerosol surface emissions and biomass burning injection heights. For trace gases all surface emissions are combined and provided as a single surface flux to the OpenIFS, with a GRIB number to identify the respective fields. For aerosol emissions these are split out between two components, which allows the introduction of a diurnal cycle to only a subset of the total emissions. Furthermore aerosol biomass burning emissions are treated separately, also allowing the application of an injection height specifically for these emissions.

4. 3D emissions of aircraft NOx emissions are provided as a separate dataset. Currently they are provided as a monthly mean dataset available on various resolutions.

5. For trace gases to which dry deposition is applied, the deposition velocities are ingested in the model similar to the emissions treatment.

If OpenIFS/AC is run in a nudging configuration then the daily (or monthly) varying surface boundary conditions (emissions and deposition) require to be provided as part of the nudging input dataset.

A standard, CAMS-based input dataset of GRIB files, as used for the model runs presented here, is provided in the distribution package along with the model code, to provide a starting point for further modification.

## 5.2 Output data

In analogy to OpenIFS, various types of output fields associated with composition-related quantities are supported by OpenIFS/AC.

1. GRIB data files containing model and/or interpolated pressure level output of user-specific tracer fields, with an output frequency of typically 3 hours. This also includes support for output of total aerosol optical depth at various wavelengths.

2. NetCDF data files containing model level output of user-specific tracer fields relying on XIOS infrastructure is available. Output settings can be configured by specific XML files.

3. Global mass and tendency diagnostics are available in the source code, allowing to analyze the evolution of tropospheric, and stratospheric burden, as well as the global, accumulated tendencies due to emission, dry and wet deposition, chemistry, as well as negative fixing and mass fixing due to non-conservation of the SL advection scheme. This capability is triggered with the `LCHEM_DIA` switch. Extended diagnostics of tendencies due to photolysis, reactions with OH can additionally be activated when setting the `LCHEM_DIAC` to true. Code infrastructure to allow output of accumulated dry and wet deposition fluxes, e.g. to study nitrogen and sulfur deposition, is in place as part of the tendency diagnostics capabilities, with deposition fluxes stored in fields `DDFLXA`





and `WDFLXA`, respectively. However, its handling is not supported in the current version of pre- and post-processing scripts.

## 5.3 Pre-, runtime- and post-processing scripts

Whereas the (Fortran-based) model code for atmospheric composition modules could essentially be taken over from the developments done as part of the CAMS activities (Flemming et al., 2015; Huijnen et al., 2016; Rémy et al., 2019), the shell scripts to be able to conveniently set up the experiments on any computing infrastructure, to run the model, and to digest model output for analysis and archiving had to be largely developed. Here we describe the main functionalities of these three aspects. Further details can be found in the `README_atmo-composition` text file provided in the code release. A `config.h` file

is used to specify the key characteristics (configuration) of the model experiment, such as the period to cover, the mode (free run / nudging), and model settings. A high-level script is available to manage a complete experiment workflow of a consistent series of consecutive OpenIFS runs. The post-processing is currently handled independently, though.

### 5.3.1 Pre-processing scripts

The pre-processing scripts prepare all the data that is needed for the requested model configuration, in particular the initial conditions and climatology. It consists of a main-level script named `prep-ic-icmcl-compo.sh`, which calls a selection of low-level scripts situated in the `/scripts/` directory. If OpenIFS is run on ECMWF computers it may access MARS archives to find necessary datasets. The pre-processing script may be run in parallel to prepare data for multiple start dates of the full OpenIFS/AC experiment. It may be set to look for already prepared datafiles instead of creating them.


### 5.3.2 Runtime scripts

The script used to actually execute an OpenIFS/AC experiment is called `oifs-run.sh.` This script reads the experiment setup from `config.h`, continues to set up a rundir (if not already done) and launches the OpenIFS executable to start or continue the experiment. A wrapper around this script is used to define the number of threads and processors, depending on

user preferences. After the experiment is finished the output data is moved over to specific directories containing the log in ASCII format, and restart files and model output files in GRIB format.

### 5.3.3 Post-processing scripts

Many downstream applications, including validation activities, require NetCDF-based model data for a selection of quantities.





Also This output data needs to be archived for later use. Such NetCDF output can be configured using the XIOS infrastructure. Another option is to select and convert the standard GRIB output from OpenIFS into the requested format using the CDO package. For this purpose a basic post-processing script is provided along with the package. This script currently supports handling of GRIB output from OpenIFS/AC, and stores this data in a user-defined location, e.g. the ECMWF's File Storage system ECFS. However, for XIOS-based NetCDF output the command structure will be very similar.

## 6 Model evaluation

### 6.1 Configurations

Here we describe the model configurations as evaluated in subsequent sections. The current configurations are:

1.  OpenIFS/AC-CB05-AER: standard configuration, using EBI solver in troposphere; OpenIFS/AC-CB05 in short.
2.  OpenIFS/AC-CB05-BASCOE-AER: configuration including stratospheric chemistry, and using the KPP solver both
in the troposphere and the stratosphere; OpenIFS/AC-CBA in short.

Both OpenIFS/AC experiments have been performed on a $T_L$ 255 spectral horizontal resolution, with 91 model levels in the vertical. The OpenIFS/AC-CB05 configuration uses chemistry table-file 'tm5ver15', which contains 56 tracers. The OpenIFS/AC-CB05-BASCOE configuration uses chemistry table-file bascoetm5ver2d.txt, and uses 100 tracers. Together with
11 tracers for the AER aerosol module, the total number of tracers in these configurations add up to 67 and 111, respectively. Initial conditions are taken on 1 January 2010 from two dedicated experiments, which are in turn based on slightly different experiments from the CAMS configuration as reported in Williams et al. (2021). The experiment using stratospheric chemistry has been initialized at altitudes above 90 hPa from a BASCOE-CTM model simulation, using the same model version and configuration as for the BASCOE Reanalysis of Aura-MLS (Errera et al., 2019) but with a finer latitude-longitude grid spacing
of 2°×2.5°. In these OpenIFS/AC experiments the vorticity and divergence (hence winds) and surface pressure are nudged towards the ERA-Interim reanalysis of meteorology (Dee et al., 2011) with a relaxation time of 5.5 hour and after vertical interpolation of the ERA-Interim data to the 91 model levels used here.

The computational costs of the different model configurations are detailed in Table 4, where we compare the OpenIFS/AC-
CB05 and OpenIFS/AC-CBA runs to experiments where only tracer transport is switched on, as well as a reference version of OpenIFS excluding the mentioned atmospheric composition components. Including the initialization and tracer transport of 67 tracers leads to a cost increase by about a factor 2.7 compared to the reference OpenIFS experiment. Increasing this number of tracers by 65% to 111 implies a further 36% cost increase. Switching on the tropospheric chemistry and aerosol processes implies a factor 3.7 increase in costs compared to the reference OpenIFS configuration, while additionally switching on the
stratospheric chemistry, and simultaneously choosing for a more expensive solver in the troposphere, results in even a factor 9.5 more expensive configuration.





Of course it should be borne in mind that the actual computational costs still depend a lot on the configuration of the model, and available computing infrastructure together with available CPUs. Other options, such as running (part of) the model in single-precision, as exploited in IFS from CY47R3 onwards, may help to reduce the costs of the model experiments. In

summary this analysis shows that switching on/off particular modules, but also for instance making deliberate choices on the outputting of fields, may result in an increase in computational costs by up to a factor 10, with, in this configuration, numbers for 'simulation years per day' ranging from 2.6 to 1.0.

**Table 4**. Computational costs for various OpenIFS/AC configurations to run a 10-day nudging experiment on $T_L255$, 91 model

levels, on 288 CPUs, using different configurations of OpenIFS and OpenIFS/AC. Costs are given in terms of runtime, and model simulation years per day (SYPD).

| Configuration | Runtime (sec) | SYPD |
|---|---|---|
| OpenIFS | 242 | 9.9 |
| OpenIFS/AC with CB05-AER tracer transport 67 tracers) | 647 | 3.7 |
| OpenIFS/AC with CB05-BASCOE-AER tracer transport (111 tracers) | 881 | 2.7 |
| OpenIFS/AC-CB05-AER, complete configuration | 911 | 2.6 |
| OpenIFS/AC-CB05-BASCOE-AER, complete configuration | 2310 | 1.0 |

## 6.2 Evaluation

In this section we describe the OpenIFS/AC model evaluation for key trace gases, viz. ozone, carbon monoxide and nitrogen

dioxide, as well as for aerosol optical depth.

### 6.2.1 Tropospheric ozone

Table 5 presents the tropospheric ozone budget for our two configurations of OpenIFS/AC. Compared to a more recent version of IFS we have a 5% larger ozone production (and loss). The methane lifetime computed by combining the tropospheric loss due to reaction with OH, with assumed loss at the surface and in the stratosphere of 70 Tg/yr (Ehhalt et al., 2001), is 9.0

(OpenIFS/AC-CBA) and 9.1 yr (OpenIFS/AC-CB05), while it is 9.9 yr in IFS CY47R1.1. These differences may be driven by differences in the OH primary production from $O_3$ photolysis, which ranges between 1677 Tg OH yr[-1] in OpenIFS and 1424 Tg OH yr[-1] in IFS CY47R1.





**Table 5**. Tropospheric ozone budget. The stratospheric inflow is calculated as the sum of the deposition and the tropospheric chemical loss minus production. Also included results from IFS CY47R1 (Williams et al., 2021) for the year 2014, for reference.

| | OpenIFS/AC-CB05 | OpenIFS/AC-CBA | IFS CY47R1 |
|---|---|---|---|
| Chemical production (Tg yr$^{-1}$) | 4762 | 4815 | 4542 |
| Stratospheric inflow (Tg yr$^{-1}$) | 348 | 336 | 247 |
| Chemical loss (Tg yr$^{-1}$) | 4148 | 4230 | 3975 |
| Dry deposition (Tg yr$^{-1}$) | 962 | 972 | 814 |
| Tropospheric burden (Tg) | 333 | 336 | 338 |
| Tropospheric lifetime (days) | 24 | 24 | 26 |

We have evaluated the tropospheric ozone mixing ratio profiles from our experiments against sonde observations from the World Ozone and Ultraviolet Radiation Data Centre (WOUDC), NOAA Earth System Research Laboratory (ESRL), and Southern Hemisphere Additional Ozonesondes (SHADOZ) networks, see Figures 1 and 2. Here we follow the regional aggregation as proposed by Tilmes et al. (2012) although a few regions have been combined, as was also done in Huijnen et al. (2019). Overall a good agreement across regions and seasons is found, with seasonal and regional mean biases within a few ppbv in the troposphere for most regional aggregates. Largest discrepancies exist for the northern hemispheric subtropical region with positive biases up to 20 ppb towards the surface, but also the model has difficulties to capture the complex chemistry over Eastern US summertime conditions, a well-known problem of the chemistry version developed in CAMS (e.g. Williams et al., 2021). Also positive biases remain in the tropical boundary layer, although it should be emphasized that the observational data coverage is very sparse with only few sonde locations.



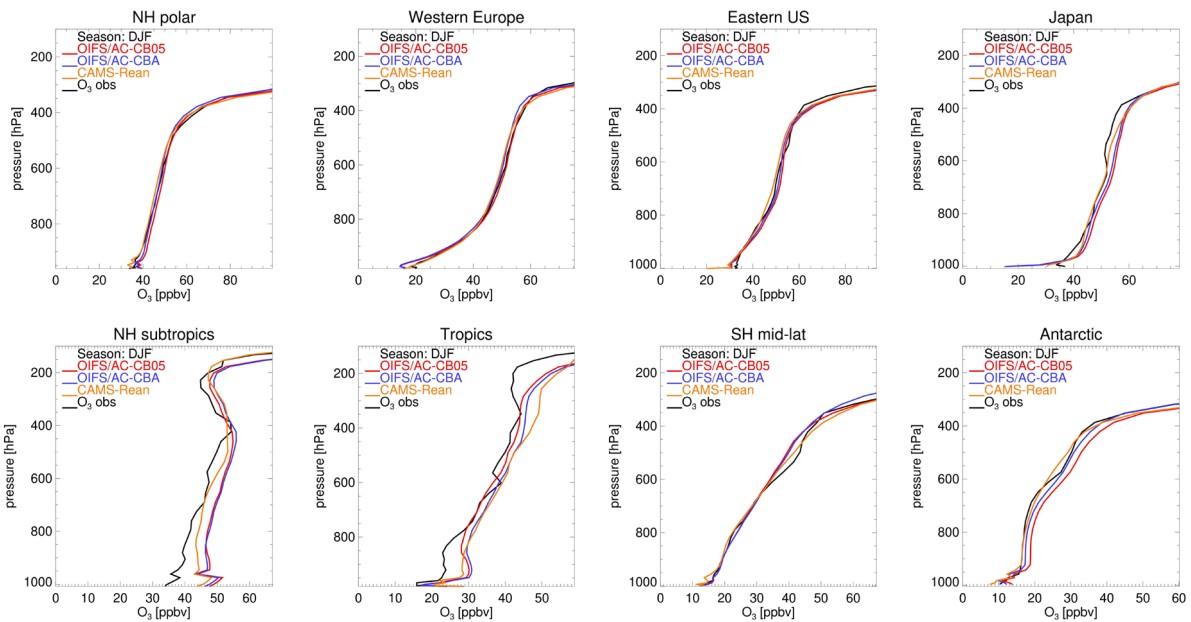

**Figure 1.** Evaluation of OpenIFS/AC tropospheric O₃ profiles against sondes during December-January-February 2010. For reference also data from the CAMS reanalysis is provided (orange).

435

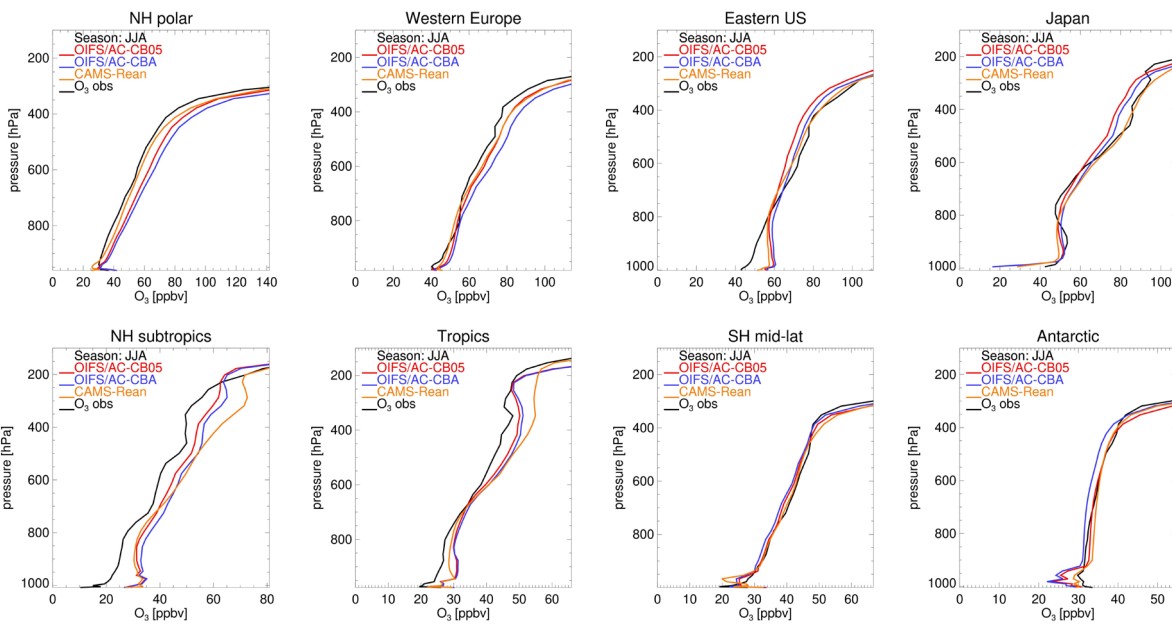

**Figure 2.** Evaluation of OpenIFS/AC tropospheric O₃ profiles against sondes during June-July-August 2010. For reference

440  also data from the CAMS reanalysis is provided (orange).

2000



### 6.2.2 Stratospheric composition

Fig. 3 shows the distributions of $O_3$, $H_2O$, $N_2O$, $HNO_3$, HCl and ClO as modelled by OpenIFS/AC-CBA on 1 October 2010
compared with chemical analyses of Aura-MLS observations by the BASCOE Data Assimilation System (Errera et al., 2019).
Fig. 4 performs the same comparisons for $CH_4$ and $NO_2$ using a BASCOE reanalysis of Envisat-MIPAS observations (Errera
et al., 2008, 2016). Overall the gas-phase composition of the stratosphere is modelled in a satisfactory manner, but some
species exhibit noticeable biases in some regions. The deficits of water vapor in the lower mesosphere and ozone in the upper
stratosphere are inherited from the BASCOE offline model (Errera et al., 2019). The latter bias is related to an overestimation
of $NO_2$ above 10 hPa (Fig. 4, bottom) and will be a target for improvement in future developments of the model.

The performance of the model in simulating polar ozone depletion is illustrated by Fig. 5 which shows decisive improvements
with the BASCOE module (OpenIFS/AC-CBA configuration) over the Cariolle parameterization (OpenIFS/AC-CB05
configuration).





**Figure 3.** Latitude-pressure distributions of zonally averaged O$_3$ , H$_2$O, N$_2$O, HNO$_3$, HCl and ClO (from top to bottom) on 1 October 2010 by OpenIFS/AC-CBA (left) and a BASCOE reanalysis of Aura-MLS observations (right).

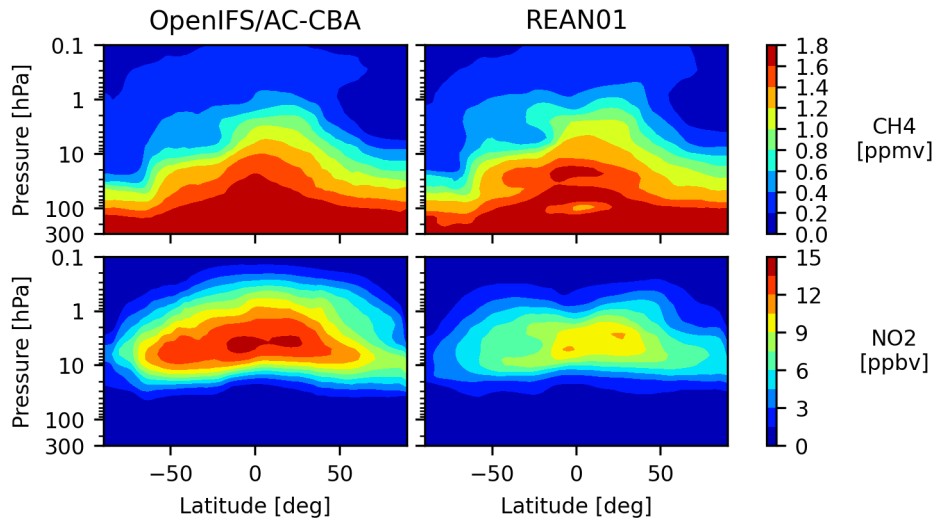

**Figure 4.** Latitude-pressure distributions of zonally averaged $CH_4$ (top) and $NO_2$ (bottom) on 1 October 2010 by OpenIFS/AC-CBA (left) and a BASCOE reanalysis of MIPAS observations (right).

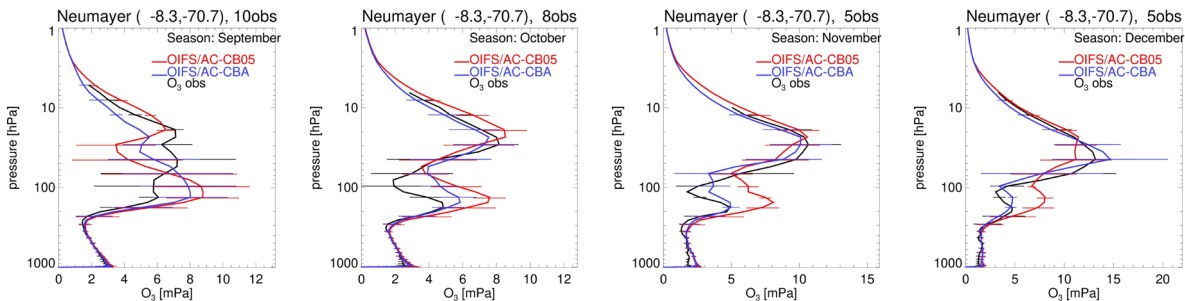

**Figure 5.** Evaluation of OpenIFS/AC stratospheric $O_3$ against sondes (black lines) during September-December 2010 at Neumayer station (8.3°W, 70.7°S) showing the OpenIFS/AC-CB05 (red lines) and the OpenIFS/AC-CBA (blue lines) configurations. Error bars represent 1-sigma variability during this period.






### 6.2.3 Tropospheric carbon monoxide

Evaluation against flask observations from the NOAA Earth System Research Laboratories (ESRL, Novelli et al., 2003), show

an overall good performance against observations, Fig. 6. Over the Northern Hemisphere during wintertime, CO mixing ratios at the surface are underestimated, which is a well-known feature in global atmospheric chemistry simulations (Shindell et al., 2006), which in part has been remedied by using larger anthropogenic emissions than originally provided in MACCity, following Stein et al. (2014). In contrast, over the Southern Hemisphere the model mostly shows a slight overestimate. Also the different start conditions between the two experiments are clearly visible, resulting in a spin-up time particularly for the

OpenIFS/AC-CB05 configuration.

This spinup effect is also visible from the CO tropospheric chemistry budget, Table 6, which shows a smaller total sink than the OpenIFS/AC-CBA configuration, associated with a smaller burden at the northern mid- and high-latitudes during the first months of 2010. Despite the larger OH, the carbon monoxide tropospheric burdens in these OpenIFS/AC configurations is also higher than those in IFS CY47R1. This is associated with higher primary emissions and secondary production of CO. This

in turn has multiple reasons. For instance, the isoprene emissions for 2010 as used in these OpenIFS/AC simulations add up to 588 Tg yr$^{-1}$, while those adopted in IFS CY47R1 are only 372 Tg yr$^{-1}$. Also the average CO yield from isoprene oxidation has decreased a bit with updated chemistry, and, e.g., due to different scavenging efficiencies for formaldehyde between the different versions of IFS.

**Table 6**. Tropospheric carbon monoxide budget. Also included results from IFS CY47R1 (Williams et al., 2021) for the year 2014, for reference.

|  | OpenIFS/AC-CB05 | OpenIFS/AC-CBA | IFS CY47R1 |
|---|---|---|---|
| Emission (Tg yr$^{-1}$) | 1037 | 1037 | 973 |
| Chemical production (Tg yr$^{-1}$) | 1580 | 1589 | 1489 |
| Chemical loss (Tg yr$^{-1}$) | 2463 | 2570 | 2468 |
| Dry deposition (Tg yr$^{-1}$) | 19 | 20 | 18 |
| Tropospheric burden (Tg) | 363 | 374 | 352 |
| Tropospheric lifetime (days) | 53 | 53 | 52 |






**Figure 6**. Evaluation of OpenIFS/AC carbon monoxide mixing ratios at the surface against NOAA ESRL flask observations.

### 6.2.3 Tropospheric nitrogen dioxide

We have performed an evaluation of tropospheric $NO_2$ columns against retrievals from OMI, using the QA4ECV product (Boersma et al., 2017), see Fig. 7. Here we have so far used 6-hourly model output, interpolated in time and space towards OMI pixels, and selecting for good-quality data, with cloud radiance fraction less than 0.5. This evaluation provides a first-order basic assessment of the model performance in terms of tropospheric $NO_2$ columns, although we acknowledge that in the future a higher model sampling frequency (at least 3-hourly) is needed. Still, this evaluation indicates that the main features





are captured by the model, showing that the large-scale spatial variation is in agreement with observations for both configurations. The evaluations suggest overall small negative biases over the continents, which is largest over Central Africa. This may be attributed to a combination of fire and soil NOx emissions. Also in this configuration of OpenIFS we do not apply a diurnal cycle to the various emission types. Smoke plumes from boreal fire emissions lead to a positive model bias.

The performance of the OpenIFS/AC-CB05 and OpenIFS/AC-CBA are mostly consistent, but large differences are seen over

the Middle East region, where the positive model bias has disappeared in OpenIFS/AC-CBA. This was found to be a consequence of the use of the KPP-based Rosenbrock solver for the tropospheric chemistry, rather than the EBI solver as adopted in the OpenIFS/AC-CB05 configuration.

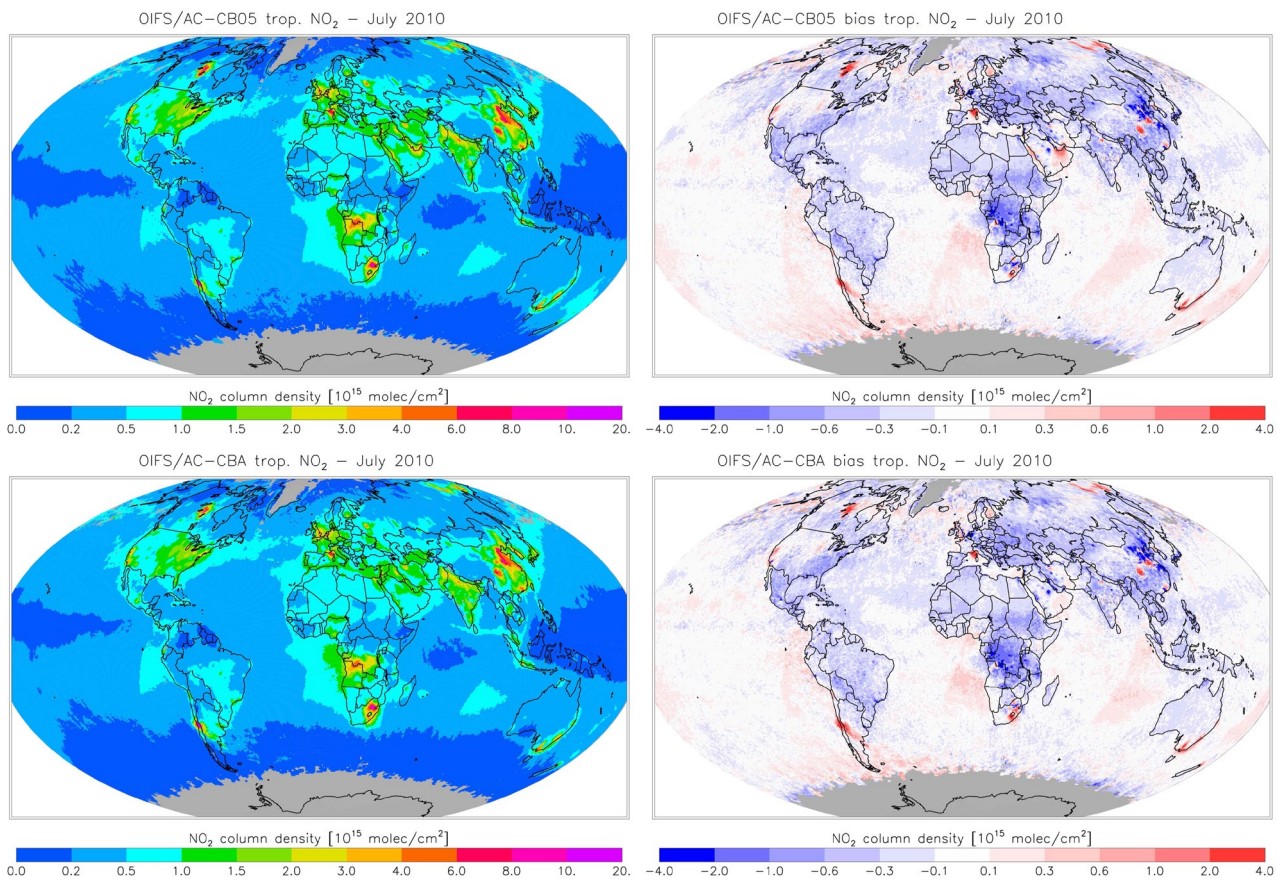

**Figure 7**. Evaluation of OpenIFS/AC-CB05 (top) and OpenIFS/AC-CBA (bottom) tropospheric $NO_2$ columns against OMI observations, averaged for July 2010. Left: model tropospheric $NO_2$ columns; right: model bias with respect to the observations.





### 6.2.3 Troposheric aerosol

Table 7 compares the simulated dust and sea-salt emissions for each bin between OpenIFS/AC-AER and different IFS versions (although for a different year) as presented in Rémy et al. (2021). Here we scale sea salt emissions and burden down by a factor 4.3 to get the dry aerosol mass. The values obtained with OpenIFS/AC can be compared to some extent with values from IFS-AER CY43R3, although a higher resolution was used ($T_L$511 spectral truncation and 137 levels over the vertical with IFS-AER against $T_L$255 and 91 levels for OpenIFS/AC). The simulated dust and sea-salt aerosols emissions are generally lower with OpenIFS/AC as compared against IFS-AER CY43R3, even though the parameterizations for these online emissions are similar between the two. This could be caused by the different resolution between the OpenIFS/AC and IFS-AER simulations: the simulated surface wind speed has been shown to be quite dependent on the time step in particular, while both sea salt aerosol and dust emissions are highly dependent on surface wind speed. Also in comparison to the more recent IFS-AER configuration in CY47R1 the sea salt emissions in OpenIFS/AC-AER are significantly lower, though compensated to some extent with a longer lifetime for bins 2 and 3.

**Table 7.** Yearly dust and dry sea salt emissions (Tg·yr$^{-1}$) / burden (Tg) / lifetime (days) respectively, as simulated by OpenIFS/AC-AER and two versions of IFS-AER. Note that both IFS-AER configurations use a higher resolution ($T_L$511/137 levels).

| Aerosol tracer | OpenIFS/AC-AER Tg·yr$^{-1}$/ Tg / days | IFS-AER CY43R3 Tg·yr$^{-1}$/ Tg / days | IFS-AER CY47R1 Tg·yr$^{-1}$/ Tg / days |
|---|---|---|---|
| Sea salt bin 1 | 36.0 / 0.09 / 0.9 | 32.2 / 0.09 / 1.0 | 110 / 0.4 / 1.3 |
| Sea salt bin 2 | 2438 / 5.2 / 0.77 | 2767 / 3.5 / 0.46 | 6596 / 4.5 / 0.25 |
| Sea salt bin 3 | 2963 / 2.1 / 0.25 | 3364 / 1.4 / 0.16 | 13658 / 1.4 / 0.04 |
| Dust bin 1 | 68 / 1.5 / 8.0 | 88 / 1.7 / 7.0 | 4.9 / 0.12 / 8.9 |
| Dust bin 2 | 181 / 3.5 / 7.0 | 292 / 5.9 / 7.2 | 45.2 / 1.0 / 8.1 |
| Dust bin 3 | 1247 / 7.2 / 2.1 | 2055 / 8.5 / 1.5 | 3248 / 13.5 / 1.5 |

We show monthly mean tropospheric AOD at 550 nm for February and July in Figure 8, and compare this to the monthly product of merged AOD by Sogacheva et al. (2020), which combines the retrievals from a wide variety of remote sensors together with AERONET data (Holben et al., 1998). The simulated and retrieved AOD show broadly the same patterns. In February, the maxima, with monthly simulated and retrieved values of 0.5–0.8 can be found over the polluted areas of China and India, as well as from biomass burning sources over Equatorial Africa. AOD over most oceans is between 0.02 and 0.1 for the simulations, while it is generally slightly higher for the retrievals. In July the dust producing regions of the Sahara, the



Middle East and Taklimakan/Gobi are also prominent, with monthly AOD values between 0.25 and 0.5 in general, for

simulation and retrieval. The patterns between simulated and retrieved AOD over dust source regions are very similar; however some underestimation is noted for transatlantic transport. Boreal fires are simulated over parts of Canada and Siberia, which correspond to retrieved values. July 2010 was also a month with exceptional fire events over Central Russia, which appears over the retrieval but less so for the simulated value. This could be caused by too low biomass burning sources of organic matter and black carbon from GFAS. Finally, over the heavily populated areas of China, Europe and the Eastern US, the simulated values are biased low as compared to the retrieved values.


**Figure 8**. Intercomparison of total AOD at 550 nm for February and July 2010, for OpenIFS/AC-CB05-AER and the FMI merged AOD product.




Figure 9 show regional comparisons of AOD at 550 nm simulated by OpenIFS/AC against AERONET data (at 500 nm wavelength), as well as against the CAMS Reanalysis (Inness et al., 2019) and the control run of the CAMS Reanalysis, which doesn't use data assimilation. In order to assess the skill of the model in terms of dust and sea-salt aerosol, results are also shown for a selection of AERONET sites that are representative of dust and sea-salt aerosol (although other aerosol types can also have some impact). The model cycle used for the CAMS Reanalysis is very close to CY43R3, which makes the control run comparable to OpenIFS/AC. AOD values from both the control run and OpenIFS/AC are generally significantly below the AERONET values, particularly over Europe, North America and over a selection of stations more representative of sea-salt aerosol. This underestimation corresponds to known issues of CY43R3 of IFS-AER, which have been improved on in later operational cycles. Over Europe and North America, a part of the underestimation comes from the fact that nitrates and ammonium are not represented. For sea-salt aerosol, the Monahan et al. (1986) scheme used in CY43R3 IFS-AER and OpenIFS/AC was shown to lead to strongly underestimated sea-salt AOD (see Rémy et al. 2021b).

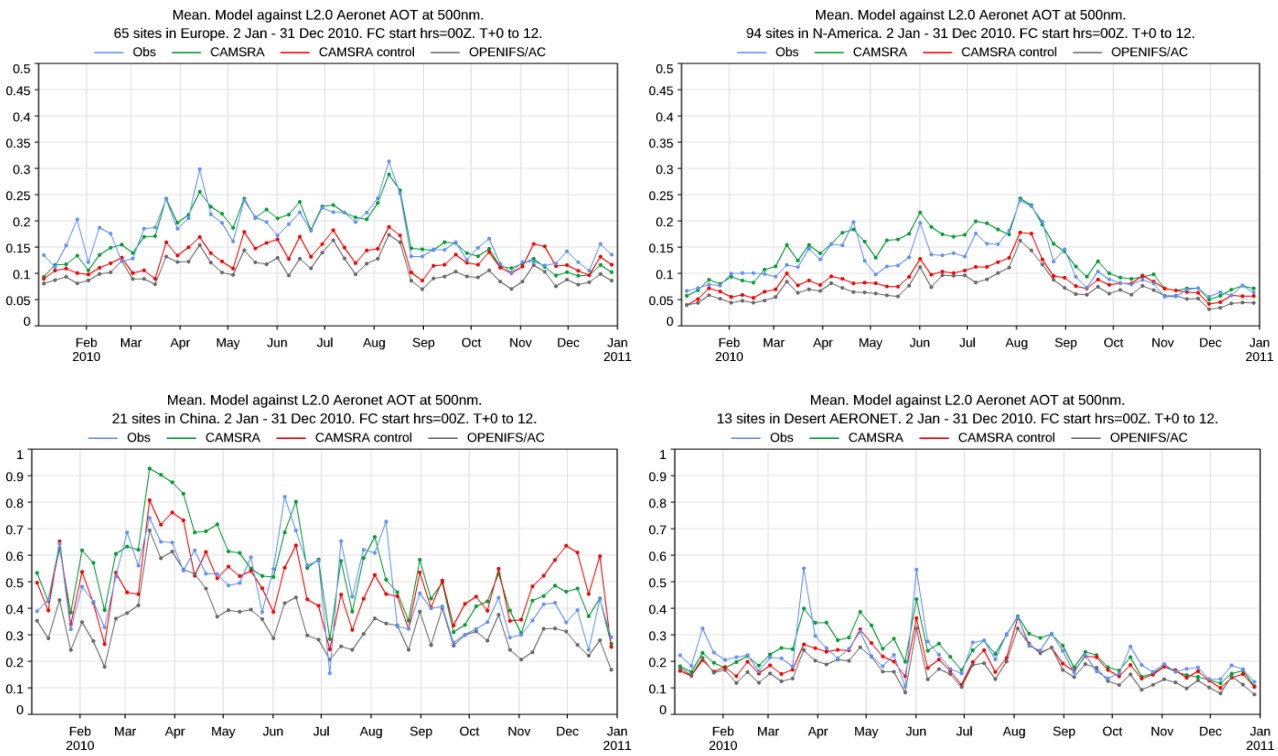





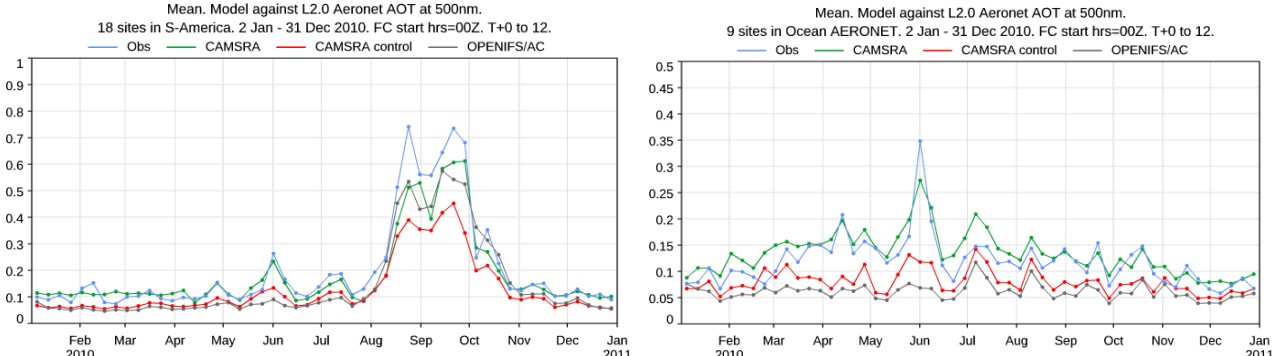

**Figure 9**. Evaluation of total AOD at 550 nm from OpenIFS/AC against observations from AERONET at 500 nm. Also results from the CAMS reanalysis, and its control experiment are included for reference.

**7 Conclusions**

We have presented a first version of OpenIFS containing atmospheric composition modules, based on OpenIFS 43r3. For this, the corresponding chemistry and aerosol modules as developed in CAMS have been introduced, along with standard input data such as emissions and dry deposition velocities. This release includes basic scripts to allow running the system in various modes, other than the nudging mode presented here. The composition model version provided is essentially the one as used

for the CAMS Reanalysis, although we emphasise that OpenIFS does not include the data-assimilation capabilities used to produce reanalysis products in CAMS. Specifically, the model contains the modified CB05 tropospheric chemistry, which can optionally be coupled to the BASCOE-based module for stratospheric chemistry, and the AER bulk-bin aerosol module.

Along with a description of the model code and input data we have provided a basic evaluation of key quantities for a one-year simulation for the year 2010, using relaxation of winds and surface pressure towards ERA-Interim data. An overall good

performance was found against various dataset for key trace gases and aerosol, e.g. showing realistic spatial variations and/or seasonal cycle in tropospheric ozone, stratospheric composition, carbon monoxide, nitrogen dioxide and aerosol optical depth.

This release makes for the first time the atmospheric composition modules integrated in ECMWF's Integrated Forecast System, as developed as part of CAMS, freely available for use by other research institutes. Also this allows the integration of alternative chemistry and aerosol schemes as the ones provided here, and allows the use of inline chemistry as part of Earth

System Modeling activities as, for instance, in EC-Earth.

Nevertheless, many of the limitations in the model configuration as presented here are known, and have already been addressed in more recent cycles of the IFS, such as for instance documented in Williams et al. (2021) and Rémy et al. (2021). With new cycles of OpenIFS coming up in future, we intend to make updated versions of the atmospheric composition module available as well. Likewise other, more up-to-date emission inventories can be expected to help address some of the biases seen here.

Finally, fundamental limitations to the existing AER bulk-bin aerosol module have motivated us to invest in the implementation of a modal scheme based on an updated version of M7, which is planned for use in EC-Earth4.

## 8. Code availability

Access to OpenIFS requires a software licensing agreement with ECMWF. OpenIFS licences are free of charge and are available to research or educational organisations. Personal licences are not provided. The use of the model and its source code

is limited to non-commercial purposes. Any operational use or the production or dissemination of real-time forecasting products is prohibited by the licence agreement. Provision of an OpenIFS software licence does not include access to ECMWF computers or data archives other than public datasets. ECMWF has limited resources to provide support and may temporarily suspend the issuing of new OpenIFS licences. Consideration may be given to requests that are judged to be beneficial for future ECMWF scientific research plans or those from scientists involved in new or existing collaborations involving ECMWF. More

details on how to access and use OpenIFS are available on the OpenIFS web portal (https://confluence.ecmwf.int/display/OIFS, last access: 8 March 2022)

OpenIFS requires a version of the ECMWF ecCodes GRIB library for input and output. ecCodes is available from the ECMWF GitHub repository (https://github.com/ecmwf, last access: 8 March 2022).

The standard configuration of OpenIFS 43r3 does not contain most of the add-ons described here (modules, scripts, input data)

which are required to model trace gases and aerosol. These can be obtained upon request as additional software to the standard model under the same licence agreement as OpenIFS itself. Parties interested in using OpenIFS/AC should therefore contact ECMWF, by emailing openifs-support@ecmwf.int, outlining their proposed use of the model.

## 9. Data availability

The OpenIFS/AC output data and scripts as used for evaluations presented in this manuscript can be downloaded from Zenodo:

(https://doi.org/10.5281/zenodo.6406674, Huijnen, 2022).

## Author contributions

VH introduced the chemistry modules into OpenIFS/AC, designed the experiments, and wrote large parts of the paper. PLS setup the runtime environment for OpenIFS/AC, and supported with code maintenance. GC and MK provided the OpenIFS code, and meteorological input. VH, SR, JF, SC and QE wrote many parts of the model code for atmospheric composition as

provided in this version of OpenIFS/AC, and contributed to the model evaluation. All authors contributed to the writing of the manuscript.



## Acknowledgements

We acknowledge the free use of tropospheric $NO_2$ column data from the OMI sensors from http://www.qa4ecv.eu (last access: 2 December 2021). We thank all research and agency teams who provided ozone sonde data to the WOUDC and SHADOZ networks. We also thank all the actors at NOAA and NASA that created and made public the CO flask observation and AERONET datasets, and the Finish Meteorological Institute (FMI) for the merged AOD product.

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
