# Peer review of "OpenIFS/AC: atmospheric chemistry and aerosol in OpenIFS 43r3"

_Geoscientific Model Development, 2022_

## Author Comment (AC1)

**Response to referee #1**

We thank the referee for her/his efforts to provide an assessment of this manuscript, and the work presented therein, and are grateful for her/his generally positive review. Below we provide a point-by-point response to the comments. The referee comments are in blue, and our response is given in black.

Summary: The paper describes in some detail the chemistry and aerosol module coupled to the OpenIFS model. The paper is well written, I don't have any major comments that would require any major revision of the paper. One comment is on the separate treatment of tropospheric and stratospheric chemistry in OpenIFS/AC in different modules, which I consider unusual and possibly worth revisiting in future releases. These two domains share a lot of important reactions (e.g. methane oxidation, NOx/HOx/Ox chemistry, CO) that would be duplicate and possibly differently represented in the two modules. Also there is a trend in tropospheric chemistry to consider halogens in the troposphere. Again there would be a lot of overlap of the tropospheric chemistry package with stratospheric halogen chemistry. So I wonder whether a single unified chemistry scheme (with perhaps options to vary aspects with this scheme, such as represent or not represent individual source gases or processes) might be the way to go. This is standard in other models. It would also eliminate the need to define an interface between the two schemes. I imagine the present approach is partly pursued because it allows for some simplification of the two separate chemistry schemes, and hence some savings in computational cost.

The referee raises a valid question regarding our choice to split the chemistry scheme into two separate parts of code, one for troposphere and one for stratosphere, instead of having a single, consistent module that encompasses the whole atmosphere. First, we should emphasize that of course the key NOx/HOx/Ox/CH4 chemistry is represented both in the troposphere and stratosphere with essentially the same chemical mechanism, and without duplications in computation of gas-phase reactions. This ensures a smooth distribution of the corresponding tracer fields across the tropopause. For other trace gases, under certain conditions, some discontinuities in mixing ratios can occur across the tropopause, particularly for short-lived trace gases that are only chemically active in one of the domains. As described in Huijnen et al. (2016), the reasons to split the algorithm into two blocks are twofold:

- It is computationally more efficient to solve chemistry mechanisms that are optimized for their target domains, e.g., solving the isoprene chemistry only in the troposphere and ozone-depleting halogen chemistry only in the stratosphere.
- Model parameterizations, in particular for photolysis and heterogeneous chemistry, have been optimized in the past for their respective domains. These cannot easily be unified without losing their respective benefits in troposphere and stratosphere.

Arguments for more unification are indeed:

- Improved consistency of the chemistry modeling, particularly the reaction rates and reaction mechanism.
- Benefits in computational costs in case of duplications, especially in the case of photolysis computations which have to be done for a full column at once.
- Improved realism, e.g., by including halogen chemistry in the troposphere and VOC chemistry in the stratosphere.

For the planning of future developments these aspects are considered. This could of course lead to model (architecture) changes in the future, similarly as for other limitations in our current model parameterizations. In the manuscript we now write in Sec. 3.2:

"The approach followed here to differentiate between tropospheric and stratospheric chemistry is chosen for reasons of computational efficiency, and combines chemical mechanisms and parameterizations (e.g., of photolysis rates and heterogeneous chemistry) that are optimized for their respective domains (Huijnen et al., 2016)."

How does the model perform at lower resolution? Many (academic) users whom this release is targeting will struggle to afford to run a model with full chemistry at TL255L91. If there are any insights around that, it would be good to comment on these qualitatively.

The model can in principle be run at many different resolutions, depending on user applications. Experience from CAMS has shown various sensitivities to model resolution, e.g. associated to NOx/VOC chemistry regimes, aerosol emissions, lightning NOx, aspects related to transport, etc., in line with widely known sensitivities reported in literature (e.g. Yu et al., 2016). Experience from CAMS has shown that the composition modeling in IFS (which has essentially the same code base as OpenIFS) behaves normally at different resolutions, and that a priori no special code modifications are needed when the resolution changes, although some tuning for dust and lightning NOx emissions may be advised.

However, running OpenIFS/AC at another resolution is not supported out-of-the-box with this release, as it requires input data on that particular resolution. Considering the basic nature of this request, guidance will be provided on how to set up the model for different resolutions. In the manuscript we now write in Sec.2:

"In the framework of CAMS, the composition modelling in the IFS has been tested at various alternative horizontal and vertical resolutions, providing scope for running OpenIFS/AC on different resolutions as well."

A little more mention of the CMIP6 model EC-Earth3-AerChem, and how the package presented here relates to this model, might be helpful. EC-Earth4 is mentioned on p3, but I'm unclear whether EC-Earth3-AerChem (which has only recently appeared in the CMIP6 archive as a late addition) is the same as the model described here, or else what the relationship is.

In EC-Earth3-AerChem, trace gases and aerosols are simulated by the TM5 chemistry and transport model. The atmospheric general circulation model of EC-Earth3-AerChem is based on IFS cycle 36r4. The two-way interaction between the two components is controlled by OASIS3-MCT. See van Noije et al. (2021) for more details. The next generation of EC-Earth (version 4) will be based on OpenIFS, and will make use of the atmospheric chemistry components presented here. As part of the ongoing development of EC-Earth4, various aspects of OpenIFS/AC, in particular pertaining to the description of aerosols and their interactions with radiation and clouds, will be revised compared to the current release.

We now refer to EC-Earth3-AerChem in the introduction and also explain in the conclusions:

"Moreover, as part of the development of EC-Earth4, the description of aerosols and their interactions with radiation and clouds will be improved. In particular, fundamental limitations to the existing AER bulk-bin aerosol module have motivated us to invest in the implementation of a modal scheme based on an updated version of M7."

The model supports two solvers: EBI and the predictor-corrector Rosenbrock solver. Ideally the choice of solver would have no discernible impact on the results. Is that indeed the case?

Experience in CAMS has indicated that there is in fact some impact of the choice of the solver on the simulation results, as well as on the computational costs. Changes appeared most prominent over conditions with large aerosol loading, associated to the solution of the chemistry involving heterogeneous reactions, in turn affecting NOx concentrations and ozone production. We now write in the manuscript, sec 3.1:

"Depending on requirements in terms of numerical accuracy, computational costs and flexibility, the user can choose between two solvers, either one based on the Euler backward iterative (EBI) method, or one based on Kinetic PreProcessor (KPP) routines, using the four stages and third-order Rosenbrock solver (Sandu and Sander, 2006)."

On the whole, subject to the minor revisions that I suggest here, I recommend publication of the paper in GMD.

Minor comments:

P1L21: Please correct citation (Naik et al., 2021?)

This has been updated to Szopa et al. (2021), thank you.

P2L34: There are examples out there where the composition model operates at a lower resolution than the meteorological model, e.g. MRI-ESM2-1. There are upsides and downsides to this approach (some cost saving while retaining small-scale dynamical features, but inconsistent transport on the two grids).

Thank you for your feedback, which are important aspects to closely assess when developing the model further.

P2L48ff: Here's where a mention of EC-Earth3-AerChem would be good to have.

EC-Earth3-AerChem is now explicitly mentioned in the Introduction.

P6L160: Offline photolysis is probably fine for the stratosphere but also likely creates inconsistencies with the approach used in the troposphere. A unified approach to photolysis would be desirable in the future, e.g. the Cloud-J scheme by UCI/ Michael Prather.

As described above there are some clear advantages for a unified photolysis scheme across the atmosphere in terms of full consistency. But this also would require considerable photolysis model development efforts if we don't want to rely on an external module completely. Also, current schemes have been proven to work well in their respective domains (Hall et al., 2018; Huijnen et al., 2016), while we note that the consistencies have been assessed and resolved where relevant. For more recent versions of stratospheric chemistry in IFS we have invested in the option to use an online scheme, which is expected to become available with updates to OpenIFS/AC. We now write:

"Later cycles of IFS also contain the option to use online computation of photolysis for the stratosphere, along with some further revisions and optimizations of this module."

P10L246: Indeed online dry deposition is preferred, e.g. using the Wesely scheme,

We agree. With newer releases of OpenIFS an online approach will become available indeed.

P11L291 ff: I admit that the filenames mean nothing to me. This level of technical detail would be best left to an operator's manual or an appendix.

This section describes the actual coding developments which is essential to operate OpenIFS/AC, and which has been developed specifically for the release of OpenIFS/AC. These scripts make this package different from the composition modeling in IFS as operated in CAMS. Therefore, even though we fully agree with the referee that it is not of scientific interest, we feel it is valuable to give this information in a separate section rather than in an appendix, to make the reader aware of such aspects. But of course this may not be of interest to the general reader. Therefore, at the start of Sec. 5 we have included a slightly more explicit disclaimer:

"In this section we describe some technical details regarding input and output data that is specific to OpenIFS/AC, to the extent this is different compared to a standard OpenIFS configuration"

Done

Done

P14L381: ERA5 has superseded ERAI. Is there a particular reason for you to use ERAI? Also, when nudging typically there would be a cut-off wavelength whereby nudging is only applied to longer wavelengths than the cutoff. Is this used here? What's the cutoff wavelength?

The referee is correct that any future application of OpenIFS/AC should really rely on more recent meteorological reanalyses, particularly ERA5. The argument for choosing ERA-Interim here was for reasons of availability of a dataset for nudging at the desired model resolution. Enabling the option for using ERA5 as an input data set for making nudged experiments at various resolutions is part of ongoing work.

Our nudging scheme doesn't apply any wavelength or spatial cutoff. Moreover, since the horizontal resolution of ERA-Interim matches with the resolution of our nudging experiments, the ERA-Interim data can be applied directly without coarsening of the data. Effectively, the impact of the nudging at the smallest spatial scales will be determined by the relaxation time (5.5 h) and the frequency with which the nudging fields are updated (every 6 h).

In the conclusions section we now write:

"…and also the application of more up-to-date meteorological input data will become available to OpenIFS/AC."

Done, thank you.

**References:**

Hall, S. R., Ullmann, K., Prather, M. J., Flynn, C. M., Murray, L. T., Fiore, A. M., Correa, G., Strode, S. A., Steenrod, S. D., Lamarque, J.-F., Guth, J., Josse, B., Flemming, J., Huijnen, V., Abraham, N. L., and Archibald, A. T.: Cloud impacts on photochemistry: building a climatology of photolysis rates from the Atmospheric Tomography mission, Atmos. Chem. Phys., 18, 16809–16828, https://doi.org/10.5194/acp-18-16809-2018, 2018.

van Noije, T., Bergman, T., Le Sager, P., O'Donnell, D., Makkonen, R., Gonçalves-Ageitos, M., Döscher, R., Fladrich, U., von Hardenberg, J., Keskinen, J.-P., Korhonen, H., Laakso, A., Myriokefalitakis, S., Ollinaho, P., Pérez García-Pando, C., Reerink, T., Schrödner, R., Wyser, K., and Yang, S.: EC-Earth3-AerChem: a global climate model with interactive aerosols and atmospheric chemistry participating in CMIP6 , Geosci. Model Dev., 14, 5637–5668, https://doi.org/10.5194/gmd-14-5637-2021, 2021.

Szopa, S., V. Naik, B. Adhikary, P. Artaxo, T. Berntsen, W.D. Collins, S. Fuzzi, L. Gallardo, A. Kiendler-Scharr, Z. Klimont, H. Liao, N. Unger, and P. Zanis, 2021: Short-Lived Climate Forcers. In Climate Change 2021: The Physical Science Basis. Contribution of Working Group I to the Sixth Assessment Report of the Intergovernmental Panel on Climate Change [Masson-Delmotte, V., P. Zhai, A. Pirani, S.L. Connors, C. Péan, S. Berger, N. Caud, Y. Chen, L. Goldfarb, M.I. Gomis, M. Huang, K. Leitzell, E. Lonnoy, J.B.R. Matthews, T.K. Maycock, T. Waterfield, O. Yelekçi, R. Yu, and B. Zhou (eds.)]. Cambridge University Press, Cambridge, United Kingdom and New York, NY, USA, pp. 817–922, doi:10.1017/9781009157896.008.

Yu, K., Jacob, D. J., Fisher, J. A., Kim, P. S., Marais, E. A., Miller, C. C., Travis, K. R., Zhu, L., Yantosca, R. M., Sulprizio, M. P., Cohen, R. C., Dibb, J. E., Fried, A., Mikoviny, T., Ryerson, T. B., Wennberg, P. O., and Wisthaler, A.: Sensitivity to grid resolution in the ability of a chemical transport model to simulate observed oxidant chemistry under high-isoprene conditions, Atmos. Chem. Phys., 16, 4369–4378, https://doi.org/10.5194/acp-16-4369-2016, 2016.

---

## Author Comment (AC2)

**Response to referee #2**

We thank the referee for her/his efforts to provide an assessment of this manuscript, and the work presented therein, and are grateful for her/his generally positive review. Below we provide a point-by-point response to the comments. The referee comments are in blue, and our response is given in black.

The paper presents a complete description of the components of a configuration of the ECMWF IFS available to the community. A rather standard evaluation of the model results are provided through comparisons to ozonesondes, satellite observations of CO, NO2, and satellite and Aeronet AOD, demonstrating reasonable performance.

Thank you for this.

The paper, and in particular the Conclusions section, rather lacks clear recommendations for the use of this model. The limitations are acknowledged, and improvements planned for future versions are mentioned, but it would be nice to see some positive statements of the value of this current version. Some recommended applications could be mentioned.

We acknowledge that the manuscript was kept compact in mentioning use cases, particularly in the conclusions section. In the introduction section we describe one of the key motivations for engaging in this work, in the framework of climate modeling, and also refer to the use of atmospheric composition modeling for the generation of satellite retrieval products. To better provide recommendations on the potential use for OpenIFS/AC, in the conclusions section we now more explicitly write:

"As such, OpenIFS/AC may foster research projects by connecting communities at the interface of meteorology, climate and atmospheric chemistry, enabling studies of trace gases and aerosols in interaction with meteorology and climate"

I think the paper is appropriate for publication in GMD.

Technical corrections:

Abstract: define OpenIFS

We now include in the abstact:

"OpenIFS is a portable version of ECMWF's global numerical weather prediction model"

l.40: define BASCOE

done

l.120: 'allows to study' should be 'allows study of' or 'allows one to study'

done, thank you

done

l.164: provide more details about the lookup table - what version of TUV was used (when were cross-section and quantum yield data updated)?

The revised text will provide the following additional details:

"Photolysis rates were computed offline by **an early version of** the TUV package (Madronich and Flocke, 1999), and are provided as lookup tables as a function of log-pressure altitude, ozone overhead column and solar zenith angle. **This version of the TUV package was originally developed for the two-dimensional model SOCRATES (Chabrillat and Fonteyn, 2003). It uses cross-section and quantum yield data from the JPL evaluation 15 (Sander et al., 2006) except for the cross-section of $Cl_2O_2$ and the quantum yields of $H_2O_2$, which were updated to the JPL evaluation 17 (Sander et al., 2011)**."

l.209-211: I found this sentence confusing - 'Following Remy et al ... as in Reddy et al.' Seems contradictory.

This reflects that Remy et al. (who describe the AER module in IFS in considerable detail) refer to Reddy et al. for this particular aspect. We now write:

"as first proposed by Reddy et al. (2005)"

**References:**

Chabrillat, S. and Fonteyn, D.: Modelling long-term changes of mesospheric temperature and chemistry, Adv. Space Res., 32, 1689–1700, doi:10.1016/S0273-1177(03)90464-9, 2003.

Sander, S., Friedl, R., Golden, D., Kurylo, M., Moortgat, G., Keller-Rudek, H., Wine, P., Ravishankara, A., Kolb, C., Molina, M., Finlayson-Pitts, B., Huie, R., and Orkin, V.: Chemical Kinetics and Photochemical Data for Use in Atmospheric Studies. Evaluation Number 15, JPL Publication 06-2, Jet Propulsion Laboratory, Pasadena, available at: http://jpldataeval.jpl.nasa.gov (last access: 31 August 2016), 2006.